# Lipoprotein(a) As a Potential Predictive Factor for Earlier Aortic Valve Replacement in Patients with Bicuspid Aortic Valve

**DOI:** 10.3390/biomedicines11071823

**Published:** 2023-06-25

**Authors:** Aleksandra Krzesińska, Maria Nowak, Agnieszka Mickiewicz, Gabriela Chyła-Danił, Agnieszka Ćwiklińska, Olga M. Koper-Lenkiewicz, Joanna Kamińska, Joanna Matowicka-Karna, Marcin Gruchała, Maciej Jankowski, Marcin Fijałkowski, Agnieszka Kuchta

**Affiliations:** 1Department of Clinical Chemistry, Medical University of Gdańsk, 80-211 Gdańsk, Poland; gabriela.chyla@gumed.edu.pl (G.C.-D.); agnieszka.cwiklinska@gumed.edu.pl (A.Ć.); maciej.jankowski@gumed.edu.pl (M.J.); agnieszka.kuchta@gumed.edu.pl (A.K.); 21st Department of Cardiology, Medical University of Gdańsk, 80-211 Gdańsk, Poland; maria.nowak@gumed.edu.pl (M.N.); agnieszka.mickiewicz@gumed.edu.pl (A.M.); marcin.gruchala@gumed.edu.pl (M.G.); marcin.fijalkowski@gumed.edu.pl (M.F.); 3Department of Clinical Laboratory Diagnostics, Medical University of Białystok, 15-269 Białystok, Poland; olga.koper@umb.edu.pl (O.M.K.-L.); joanna.kaminska@umb.edu.pl (J.K.); joanna.matowicka-karna@umb.edu.pl (J.M.-K.)

**Keywords:** lipoprotein(a), Lp(a), bicuspid aortic valve, BAV, aortic valve stenosis, AVS, cardiovascular disease, lipid profile parameters, autotaxin

## Abstract

Bicuspid aortic valve (BAV) affects 0.5–2% of the general population and constitutes the major cause of severe aortic valve stenosis (AVS) in individuals ≤70 years. The aim of the present study was to evaluate the parameters that may provide information about the risk of AVS developing in BAV patients, with particular emphasis on lipoprotein(a) (Lp(a)), which is a well-recognized risk factor for stenosis in the general population. We also analyzed the impact of autotaxin (ATX) and interleukin-6 (IL-6) as parameters potentially related to the pathomechanism of Lp(a) action. We found that high Lp(a) levels (>50 mg/dL) occurred significantly more frequently in patients with AVS than in patients without AVS, both in the group below and above 45 years of age (*p* = 0.036 and *p* = 0.033, respectively). Elevated Lp(a) levels were also strictly associated with the need for aortic valve replacement (AVR) at a younger age (*p* = 0.016). However, the Lp(a) concentration did not differ significantly between patients with and without AVS. Similarly, we observed no differences in ATX between the analyzed patient groups, and both ATX activity and concentration correlated significantly with Lp(a) level (R = 0.465, *p* < 0.001 and R = 0.599, *p* < 0.001, respectively). We revealed a significantly higher concentration of IL-6 in young patients with AVS. However, this observation was not confirmed in the group of patients over 45 years of age. We also did not observe a significant correlation between IL-6 and Lp(a) or between CRP and Lp(a) in any of the analyzed groups of BAV patients. Our results demonstrate that a high level of Lp(a), greater than 50 mg/dL, may be a significant predictive factor for earlier AVR. Lp(a)-related parameters, such as ATX and IL-6, may be valuable in providing information about the additional cardiovascular risks associated with developing AVS.

## 1. Introduction

Bicuspid aortic valve (BAV) is a congenital anomaly resulting from the fusion of the aortic valve cusps during prenatal development, causing the valve to have only two cusps instead of three. It represents the most common valvular congenital defect, with a prevalence of 0.5–2% in the general population [1,2,3]. One of the most frequently diagnosed complications of BAV is aortic valve stenosis (AVS), which occurs at a relatively younger age compared to patients with a tricuspid aortic valve (TAV) [4,5]. The pathophysiology of AVS is divided into an initiation phase characterized by lipid infiltration, oxidation, and inflammation and a propagation phase including fibrosis and calcification of aortic cusps. BAV is the major cause of AVS in individuals ≤70 years and leads to a more severe course of AVS [6,7,8,9]. There is no medical treatment to reduce the symptoms or to stop or delay the progression of AVS, either in TAV patients or BAV patients. Currently, the only available and effective intervention leading to better survival outcomes is surgical or transcatheter aortic valve replacement. It is estimated that about 50% of patients with clinical symptoms of AVS die within the next 12–18 months if surgical treatment is not applied [10,11,12,13]. The main risk factors for AVS in the general population are shared with the main risk factors for atherosclerotic cardiovascular disease (ASCVD) and include age, obesity, lipid disturbances, hypertension, and smoking habits [10,14]. In BAV patients, apart from continuous shear stress, wear, and tear of the aortic valve, the importance of other risk factors in the development and progression of AVS remains unclear. Recently, particular interest has focused on lipoprotein(a) (Lp(a)), which is an important, causal, and nonmodifiable predictor of coronary artery disease (CAD) and valvular outcomes in the overall population [14,15]. 

Lp(a) is a cholesterol-rich particle containing the large glycoprotein, apolipoprotein(a) (apo(a)), covalently attached to apolipoprotein B-100 (apo B-100) via a single disulfide bond. Genetic background is the major factor determining circulating serum Lp(a) levels, through the *LPA* gene encoding apo(a). Apo(a) shares significant sequence similarity with plasminogen and consists of the kringle IV repeat domain (KIV), the kringle V domain, and a protease-like domain. Only KIV2 is repeated in the apo(a) sequence, and the number of KIV2 repeats gives rise to the high size heterogeneity in apo(a) isoforms and is inversely correlated with the plasma concentration of Lp(a) [16]. The pathophysiological mechanism of Lp(a) action is still under investigation. It has been claimed that Lp(a) may deliver cholesterol to valve leaflets and mediate the formation of microcrystals, which are a potential source of calcification [14,17]. Lp(a) is also the main carrier of oxidized phospholipids (OxPLs), which lead to increased oxidative stress and inflammation—common pathomechanisms of vascular endothelial dysfunction and the initial stages of valve osteogenesis. OxPLs are also involved in the process of mineralization of vascular cells by signaling bone morphogenic proteins and upregulation of osteoblastic transcription factors [18,19,20]. Additionally, the *LPA* gene contains an interleukin-6 (IL-6) response element. Therefore, IL-6 can upregulate Lp(a) production, although it appears that Lp(a) induces the expression of inflammatory cytokines, including IL-6 [21,22]. Autotaxin (ATX), another molecule associated with Lp(a), also contributes to valve calcification, fibrosis, and inflammation. Some clinical studies have confirmed that Lp(a) may transport ATX to aortic valve damage and have shown that ATX is an important component of Lp(a) toxicity [23,24]. 

Due to the high risk of developing AVS in patients with BAV, the well-established significance of high Lp(a) levels in AVS, and recent reports that provide hope for effective pharmacotherapy of elevated levels of Lp(a) [25], the aim of the project is to determine whether the concentration of Lp(a) may be a potential risk marker useful for stratification of people at higher risk of developing AVS in the BAV population. We also analyzed the impact of ATX and IL-6 as parameters potentially related to the pathomechanism of Lp(a) action.

## 2. Materials and Methods

### 2.1. Patients

The study group included 75 patients with BAV, aged 27–78 years, from the 1st Clinic of Cardiology and the General Practitioner Clinic at the Medical University of Gdańsk (Poland). Among them, there were 44 individuals with AVS and 31 without AVS. Two age groups were distinguished in the study population below and above 45 years of age based on the fact that in the case of atherosclerosis, diagnosis of the disease in patients younger than 45 years of age is termed as young CAD [26,27].

The primary inclusion criterion for the study was a diagnosis of BAV confirmed by echocardiography. The criteria for exclusion from the study were as follows: heart failure at baseline, severe clinical condition of the patient, active inflammatory process measured by C-reactive protein (CRP) (concentration > 10.0 mg/L) and morphology, untreated hypothyroidism, severe chronic kidney failure (glomerular filtration rate < 30 mL/min/1.73 m^2^), administration of anti-inflammatory or immunosuppressive drugs, and pregnancy. Clinical data regarding age, family history of cardiovascular diseases (CVDs), coexistence of CAD, arterial hypertension, diabetes, chronic kidney disease, obesity, dyslipidemia, smoking status, and medications taken were collected via a questionnaire completed by the patient. According to the American Heart Association, CVDs were considered to be a group of disorders of the heart and blood vessels and included heart disease, heart attack, stroke, heart failure, arrhythmia, and also heart valve problems.

The study was conducted in accordance with the ethical guidelines of the 1975 Declaration of Helsinki and was approved by the Independent Bioethics Committee for Scientific Research at the Medical University of Gdansk (NKBBN/28/2022). All of the patients provided informed consent.

### 2.2. Echocardiography

Transthoracic echocardiography was performed on each patient. Aortic valve morphology was evaluated. AVS was analyzed by measuring aortic valve peak transvalvular velocity (V_max_), mean transvalvular pressure (PG_mean_), and aortic valve area (AVA). The left ventricular ejection fraction (LVEF), the presence of aortic regurgitation (AVR), and the dimensions of the aortic complex were assessed. 

### 2.3. Laboratory Measurements

Peripheral fasting blood samples were drawn between 7 a.m. and 8 a.m. following the overnight fast. The serum was separated by centrifugation at 1000× *g* for 15 min and was stored at −80 °C pending analysis. Total cholesterol (TC), high-density lipoprotein cholesterol (HDL-C), and triglycerides (TG) were determined in serum using standard enzymatic colorimetric tests (Wiener Lab, Warsaw, Poland). Low-density lipoprotein cholesterol (LDL-C) was calculated using the Friedewald formula. The apolipoprotein AI (Apo AI) and Apo B serum concentrations were measured using the nephelometric method with antibodies obtained from Siemens Healthcare Diagnostics (Eschborn, Germany) on a Behring laser nephelometer. The Lp(a) concentration was measured using a commercially available immunoturbidimetric assay (Randox, Crumlin, UK). ATX activity was determined based on a hydrolysis rate of 1 myristoyl-sn-glycero-3-phosphocholine by measuring choline as a reaction product by the colorimetric assay (Merck, Warsaw, Poland) [28]. The total ATX concentration was analyzed in serum using enzyme immunoassay kits (Biorbyt, Cambridge, UK). The COBAS e411 immunochemical analyzer (Roche Diagnostics, Warsaw, Poland), which uses an IVD (in vitro diagnostic) test with the electrochemiluminescence immunoassay (ECLIA) method, was used to measure the concentration of serum IL-6.

### 2.4. Statistical Analysis 

Statistical analyses were performed using STATISTICA software version 10 (StatSoft, Warsaw, Poland) and GraphPad Prism 5.0 (GraphPad Software, San Diego, CA, USA). A Shapiro–Wilk test was used to assess the normality of the distribution of the variables. The continuous variables were expressed as medians with 25th and 75th percentiles. Data analysis was performed using the nonparametric Mann–Whitney U test. Pearson’s chi-square test was used to compare categorical variables, and Spearman’s standardized coefficients were used to assess univariate correlations. Surgery-free survival analysis was determined based on a Kaplan–Meier analysis, and the comparison of survival curves was made using the Gehan–Wilcoxon test. A *p*-value below 0.05 was considered statistically significant.

## 3. Results

Baseline clinical characteristics, echocardiographic evaluation, and cardiovascular risk parameters of the study population are shown in Table 1 and Table 2. The study population was divided into two age groups, namely, below and above 45 years of age, with respect to the presence of AVS.

The lipid profile parameter concentrations, with the exception of Apo B, were not significantly different between the study groups of BAV patients with and without AVS (Table 3). The Apo B concentration was significantly higher in subjects with AVS but only in the group below 45 years of age. The differences shown in Lp(a) concentrations were not significant (Figure 1); however, BAV patients with AVS had significantly more frequently higher levels of Lp(a). In the group of younger patients, 56% of patients with AVS had a high level of Lp(a) (>50 mg/dL), while in the group without stenosis, only one patient (7%) had a high level of Lp(a). In the group of older patients, over 90% of patients without AVS had low levels of Lp(a) (<50 mg/dL), and only one patient in this group had an increased concentration of Lp(a) (Table 3). We observed a significantly higher level of IL-6 in young patients with AVS. However, this observation was not confirmed in the group of patients over 45 years of age. We also found no significant correlation between IL-6 and Lp(a) or between CRP and Lp(a) in any of the analyzed groups of BAV patients. The ATX concentration and activity were not significantly different between the groups with respect to AVS. In the entire study population, ATX concentration and activity correlated significantly with Lp(a) (Figure 2). 

In the studied BAV patients, the need for surgical replacement of the valve occurred in 28 patients; however, among the analyzed variables (concentration of IL-6, ATX, Lp(a), lipid profile parameters, diabetes, smoking, Lp(a) > 30 mg/dL, and Lp(a) > 50 mg/dL), only a high concentration of Lp(a) with a cut off 50 mg/dL was predictive of a shorter surgical-free survival (*p* = 0.016) (Figure 3).

## 4. Discussion

In our study, we showed that an elevated level of Lp(a) (>50 mg/dL) plays an important role in the direction of the course of AVS in BAV patients and may be a significant predictive factor for the need for AVR at a younger age. We estimated that 50% of patients with BAV and a high Lp(a) concentration underwent surgical valve replacement before the age of 62 years, while among patients with AVS and a low Lp(a) concentration, half of the patients had a chance of survival until the age of 69 without surgical intervention. Our data are in line with the results of the prospective Astronomer trial involving patients with mild to moderate AVS, indicating the coexistence of increased Lp(a) levels with faster progression rates and the need for surgical treatment in these individuals [29].

Elevated serum Lp(a) levels (above 30–50 mg/dL) are a well-recognized risk factor for ASCVD in the general population [25]. It has been claimed that the small isoform of apo(a) associated with Lp(a) levels > 50 mg/dL increases ASCVD risk by 2–2.5-fold compared to the large apo(a) isoform [15,30]. Nowadays, there is a convincing body of evidence indicating that Lp(a) is also an independent and likely causal risk factor for the presence and progression of AVS in the general population [14]. One of the earliest works noting the association between Lp(a) and AVS was the Gotoh et al. study that included 784 participants aged ≥35 years; it showed higher incidence of AVS in patients with elevated Lp(a) levels, even with the cut-off point of 30 mg/dL [31]. Subsequently, several other population-based studies reported similar relationships, including a cohort MESA study, which indicated that a high Lp(a) concentration was combined with a higher degree of calcification in White and Black participants [32,33,34,35,36]. In a prospective population-based analysis, Arsenault et al. reported that Lp(a) ≥ 50 mg/dL was associated with significantly increased AVS risk over a follow-up of 11.7 years [34].

The impact of Lp(a) on the risk of AVS in BAV patients has been much less elucidated. However, a few recent studies have indicated that Lp(a) may also be an important factor modulating the progression of BAV to AVS. Sticchi et al. observed a significantly increased serum Lp(a) concentration according to the degree of calcification in a group of 69 patients with AVS. The study also assessed the number of *LPA* KIV2 repeats, which determine the size of apo(a) and are inversely correlated with the serum concentration of Lp(a). The results showed that despite no significant relationship between KIV2 repeat quantity and the serum level of Lp(a) with BAV per se, subjects with a more severe degree of calcification had lower *LPA* KIV2 repeat numbers, parallel to the elevated Lp(a) level [37]. Another small case series study by Ker et al. performed on 10 patients showed a clear association between serum Lp(a) concentrations and the presence or absence of calcification in BAV [38]. Our results showed that a high level of Lp(a) is not only a predictor of the AVS course but also occurs significantly more often in the group of patients with BAV stenosis compared to the group without symptoms of stenosis, which is consistent with the above-mentioned observations and confirms the significance of Lp(a) levels in AVS of BAV individuals as a concept worthy of attention and exploring.

In parallel with the serum concentration of Lp(a) in the context of AVS, an interesting aspect seems to be the study of Lp(a)-related parameters, which may be connected with the pro-calcification mechanisms of Lp(a). ATX is one such molecule, preferentially transported by Lp(a), which hydrolyzes lysophosphatidylcholine to lysophosphatidic acid, a bioactive phospholipid that participates in valve mineralization, fibrosis, and the inflammatory response [24]. In a study by Torzewski et al., the immunochemistry of aortic valve leaflet lesions from 68 patients with different grades of AVS revealed the presence of ATX in the valve leaflet, particularly in advanced stages of calcification [39]. Nsaibia et al. compared patients with CAD with and without concomitant calcification in stenosis and showed that an increased risk of calcification in AVS was associated with elevated ATX activity and ATX mass [40]. In our work, we determined both the serum concentration and the activity of ATX in all patients with BAV. We did not observe a significant association between the values of these parameters and AVS. However, both ATX activity and concentration showed significant correlations with Lp(a), which seems to confirm the findings obtained by Bourgeois et al. that ATX is strictly associated with Lp(a) and shows its potential clinical significance as worthy of further investigation [23].

Similar to the pathomechanism of atherosclerosis, many studies have pointed to the importance of pro-inflammatory processes in the pathomechanism of AVS development. CRP and IL-6, as markers of inflammation, have been widely studied and described as factors in the development of atherosclerosis. Some works found a positive association between levels of plasma IL-6 and the progression of AVS. Furthermore, not only the concentration of IL-6 but also the genetic variants of this parameter and its receptor seem to be important in the clinical status of AVS [41,42]. The *LPA* gene contains an IL-6 response element [43], and some data support a positive association between IL-6 and Lp(a). It has been claimed that Lp(a) levels are elevated in patients with increased serum IL-6, which is most closely related to an increased risk of atherosclerosis [22]. The link between CRP and AVS development is not clear. Some researchers have suggested no association [44,45], but others have suggested an association between CRP levels and the incidence of AVS even in a population with low CRP levels [46]. In the study by Topçiu-Shufta et al., evaluating a correlation of inflammation with Lp(a) in hemodialysis patients, significant positive correlations were observed in individuals with high CRP levels (>10 mg/L), not only between Lp(a) and CRP but also between Lp(a) and IL-6 concentration [47]. In our study, we showed no significant correlation between either IL-6 and Lp(a) or between CRP and Lp(a) in any of the analyzed groups of BAV patients; however, our study did not include patients with acute inflammation (CRP > 10 mg/L) according to the exclusion criteria. Nevertheless, we demonstrated a significantly higher concentration of IL-6 in patients below 45 years of age with AVS, which may suggest an incipient inflammatory process, one of the common pathomechanisms of valvular anomaly [21]. However, this observation was not confirmed in the group of older patients, which may have resulted from a statistically significant difference in statin therapy (Table 1), as well as from the fact that with increasing age, additional other risk factors have an impact on inflammation.

The prevalence of dyslipidemia in patients with BAV has still not been established, and the researchers’ findings were equivocal. We observed no significant differences in any of the traditional lipid profile parameters, including TC, LDL-C, HDL-C, and TG, in either age group with respect to AVS. In the young individuals (<45 years of age), Apo B concentration was significantly increased in subjects with AVS than without AVS, while among patients above 45 years of age, no significant differences were present, which may result from the fact that there were more patients on statin therapy in this group. Mundal et al. showed a remarkably increased prevalence of AVS in subjects with familial hypercholesterolemia [9]. Moreover, a case-control study by Gracia Baena et al. revealed that dyslipidemia was one of the cardiovascular risk factors associated with an increased risk of AVS [48]. However, there is also research indicating an insignificant relationship between TC, LDL-C, and TG and cross-sectional aortic diameters in individuals with BAV who underwent AVR [49]. Our results seem to be in agreement with the study of Alegret et al., which suggested that the concentration of Apo B could predict ascending aorta diameter in BAV patients [50]. Considering the importance of Apo B as an indicator of cardiovascular risk [51] and the equivocal test results mentioned above, future clinical trials should evaluate dyslipidemic-related serum biomarkers in the context of BAV-induced AVS for a relatively personalized diagnosis and stratification.

Limitations of this study include the relatively small sample size and the lack of assessment of the degree of calcification by computed tomography imaging. The sample size was partly a consequence of the number of subjects who were under the care of our center, the number of patients who agreed to participate in the studies, and the exclusion criteria that influenced the final number of the entire BAV population.

## 5. Conclusions

Our results demonstrated that an elevated level of Lp(a), greater than 50 mg/dL, may be an important factor characterizing the course of BAV and may be a significant predictive factor for earlier AVR. Lp(a)-related parameters, such as ATX and IL-6, may provide noteworthy details about additional ASCVD risk components and the risk of developing AVS. Further investigations are essential to clarify the detailed pathomechanisms of Lp(a) action connected with ATX and IL-6 in the course of BAV and to predict cardiovascular complications in patients at increased risk of developing a severe course of AVS.

## Figures and Tables

**Figure 1 biomedicines-11-01823-f001:**
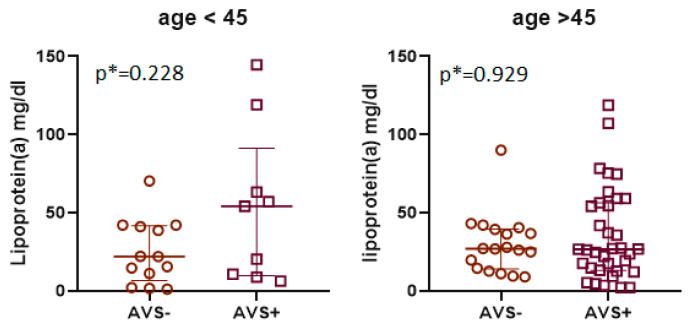
Lipoprotein(a) concentration in BAV patients with and without AVS. Values are presented as single points (dots for patients without AVS and squares for patients with AVS) and medians with interquartile ranges. Values were assessed using the Mann–Whitney U test (*).

**Figure 2 biomedicines-11-01823-f002:**
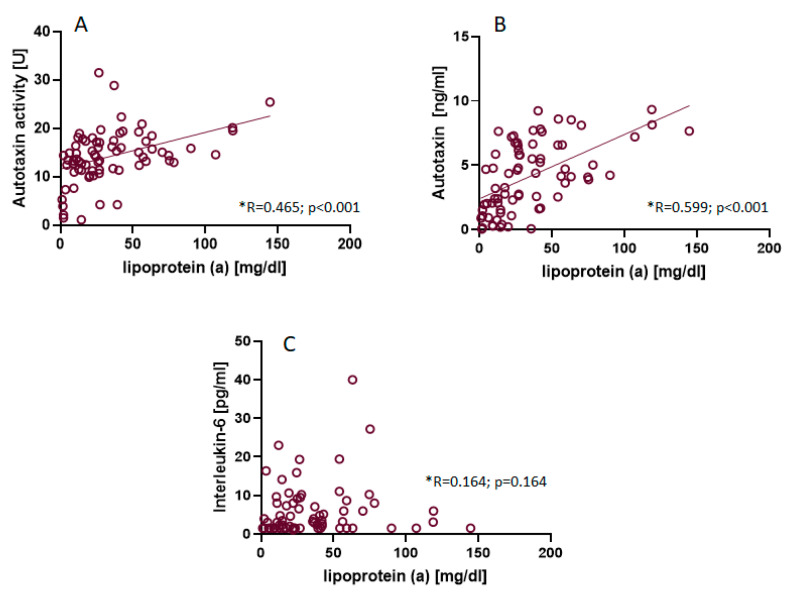
Univariate correlation between lipoprotein(a) concentration and autotaxin activity (**A**), autotaxin concentration (**B**), and interleukin-6 concentration (**C**). Values are presented as single points and expressed as standardized Spearman coefficients (*).

**Figure 3 biomedicines-11-01823-f003:**
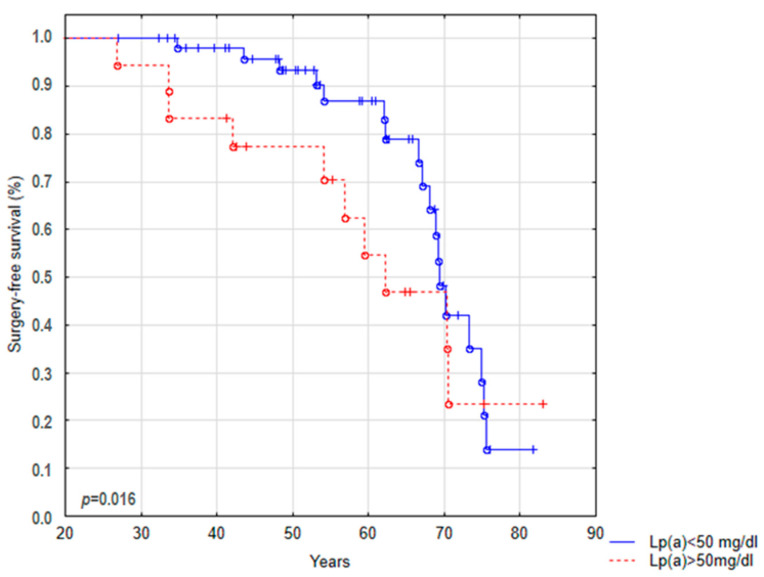
Kaplan–Meier curve for the surgery-free survival analysis of the entire BAV population. The comparison of survival curves was made using the Gehan–Wilcoxon test.

**Table 1 biomedicines-11-01823-t001:** Baseline clinical characteristics.

	All Patients	Age < 45		*p*	Age > 45		*p*
		AVS–	AVS+		AVS–	AVS+	
n	75	13	9		18	35	
Age, years	54 (44–65)	39 (31–41)	40 (34–43)	0.367	55 (49–61)	65 (59–69)	0.002
BMI, kg/m^2^	27 (24–30)	27 (24–32)	24 (23–27)	0.190	27 (25–30)	28 (26–31)	0.247
Sex (male), N (%)	21 (28%)	2 (11%)	4 (44%)	0.132	3 (16%)	12 (35%)	0.147
Smoking habits, N (%)	24 (42%)	3 (23%)	2 (22%)	0.919	7 (39%)	12 (35%)	0.962
Hypertension, N (%)	39 (52%)	2 (15%)	1 (11%)	0.854	13 (72%)	23 (65%)	0.795
Diabetes, N (%)	9 (15%)	1 (1%)	0	0.421	2 (11%)	6 (17%)	0.434
Statins therapy, N (%)	31 (41%)	3 (23%)	1 (11%)	0.435	5 (27%)	22 (62)	0.010
CAD, N (%)	21 (28%)	0	1 (12%)	0.191	6 (33%)	14 (40%)	0.518
Family history of CVD, N (%)	37 (49%)	6 (46%)	6 (67%)	0.342	10 (55%)	15 (43%)	0.345

Values are presented as medians (25th and 75th percentiles). Data analysis was performed using the nonparametric Mann–Whitney U test and Pearson’s chi-square test. AVS—aortic valve stenosis; BMI—body mass index; CAD—coronary artery disease; CVD—cardiovascular disease.

**Table 2 biomedicines-11-01823-t002:** Echocardiographic evaluation of patients in each age group.

	Age < 45			Age > 45		
	AVS–	AVS+	*p*	AVS–	AVS+	*p*
LVEF (%)	60 (40–64)	53 (32–70)	0.965	60 (55–64)	60 (51–64)	0.965
AVR, N (%)	10 (76%)	7 (77%)	0.7	15 (83%)	14 (40%)	0.18
Aortic aneurysm, N (%)	8 (61%)	5 (55%)	0.8	17 (94%)	20 (57%)	0.35
AVA (cm^2^)	2.4 (2.1–2.5)	1.3 (0.9–1.4)	<0.001	2.5 (2.2–2.6)	0.9 (0.8–1.2)	<0.001
V_max_ (m/s)	1.7 (1.6–2)	3.6 (2.4–4.0)	<0.001	1.7 (1.5–1.8)	3.9 (3.4–4.4)	<0.001
PG_mean_ (mmHg)	7 (6–14)	32 (15–36)	<0.001	7 (6.5–8)	40.5 (25–46.5)	<0.001

Values are presented as medians (25th and 75th percentiles). Data analysis was performed using the nonparametric Mann–Whitney U test and Pearson’s chi-square test. AVA—aortic valve area; V_max_—aortic valve peak transvalvular velocity; PG_mean_—mean transvalvular pressure gradient; LVEF—left ventricular ejection fraction; AVR—aortic valve regurgitation.

**Table 3 biomedicines-11-01823-t003:** Lipid profile and Lp(a)-related parameters.

	All Patients	Age < 45			Age > 45		
		AVS–	AVS+	*p*	AVS–	AVS+	*p*
TC [mg/dL]	140 (120–159)	151 (139–166)	150 (126–164)	0.441	144 (124–167)	128 (113–147)	0.065
HDL-C [mg/dL]	51 (46–58)	52 (47–55)	48 (45–52)	0.705	51 (46–60)	50 (42–58)	0.428
LDL-C [mg/dL]	99 (83–108)	101 (997–114)	107 (94–110)	0.449	99 (91–119)	86 (79–105)	0.060
TG [mg/dL]	90 (61–123)	83 (47–113)	63 (52–108)	0.659	105 (69–138)	87 (68–113)	0.272
Apo B [mg/dL]	98 (81–110)	95 (77–100)	110 (102–132)	0.035	100 (93–114)	83 (77–105)	0.087
Apo AI [mg/dL]	152 (139–173)	150 (126–168)	143 (131–151)	0.395	167 (139–183)	155 (146–173)	0.655
Lp(a) [mg/dL]	26 (12–47)	22 (6.5–41)	54 (10–91)	0.228	27 (14–40)	26 (13–56)	0.929
Lp(a) < 50 mg/dL [n(%)]	58 (76%)	12 (93%)	4 (44%)	0.036	17 (94%)	24 (69%)	0.033
Lp(a) > 50 mg/dL [n(%)]	18 (24%)	1 (7%)	5 (56%)		1 (6%)	11 (31%)	
CRP [mg/L]	1.17 (0.5–2.6)	1.2 (0.9–2.9)	1.2 (0.4–2.2)	0.656	1.0 (0.9–6.7)	1.2 (0.4–2.2)	0.373
IL-6 [pg/mL]	3 (1.5–8)	1.5 (1.5–2.7)	5.9 (1.5–9.7)	0.043	3.4 (1.5–8.6)	3.5 (1.5–8.7)	0.710
ATX [U]	14 (11–17)	15 (13–18)	16 (14–19)	0.455	13 (11–17)	13 (12–16)	0.875
ATX [ng/mL]	4.2 (1.9–6.5)	1.6 (1.1–5.6)	4.3 (2.3–6.5)	0.141	3.4 (1.5–8.6)	4.4 (2.4–6.7)	0.962

Values are presented as medians (25th and 75th percentiles). Data analysis was performed using the nonparametric Mann–Whitney U test and Pearson’s chi-square test. TC—total cholesterol; HDL-C—high-density lipoprotein cholesterol; LDL-C—low-density lipoprotein cholesterol; TG—triglycerides; Apo B—apolipoprotein B; Apo AI—apolipoprotein A-I; Lp(a)—lipoprotein(a); CRP—C-reactive protein; IL-6—interleukin-6; ATX—autotaxin.

## Data Availability

The data presented in this study are available upon request from the corresponding author.

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
