# Peer review of "Lipoprotein(a) As a Potential Predictive Factor for Earlier Aortic Valve Replacement in Patients with Bicuspid Aortic Valve"

_biomedicines, 2023, doi:10.3390/biomedicines11071823_

Round 1

Reviewer 1 Report

The work is devoted to the study of increased concentration of Lp(a) as a marker of aortic valve replacement. Determination of Lp(a) concentration in people with congenital valve pathology may allow timely identification of high-risk patients, given that the concentration of Lp(a) is determined genetically and practically does not change throughout life. The small number of included patients slightly reduces the significance of the results, but the authors understand this and describe this as a limitation of their study. The manuscript is easy to read, all sections of the manuscript are described clearly and in sufficient detail. During the review process, however, several questions and wishes arose: 1.Although the authors write that "Clinical data regarding age, family history of cardiovascular disease, coexistence of CAD, arterial hypertension, diabetes, chronic kidney disease, obesity, dyslipidemia, smoking status, and medications taken were collected via a questionnaire completed by the patient" , data on family history of cardiovascular disease and coexistence of CAD are not available in Table 1. Given that elevated Lp(a) is a risk factor for early ASCVD, this information should be added. 2. Line 134-135. It is not entirely clear what parameter the description “An IVD (in vitro diagnostics) test was utilized, which employed the electrochemiluminescence immunoassay (ECLIA) method” refers to. 3. In table 1 the  dimension of  BMI should be added. 4. I recommend removing the "*" in the tables, you can leave only the description of the statistical methods in captions under the tables. 

Author Response

Thank you very much for your review. We very appreciate the important and valid comments. We hope that we have managed to respond to all the comments in a manner that the reviewer will find satisfactory.

Response to Reviewer 1 Comments

Point 1: Although the authors write that "Clinical data regarding age, family history of cardiovascular disease, coexistence of CAD, arterial hypertension, diabetes, chronic kidney disease, obesity, dyslipidemia, smoking status, and medications taken were collected via a questionnaire completed by the patient" , data on family history of cardiovascular disease and coexistence of CAD are not available in Table 1. Given that elevated Lp(a) is a risk factor for early ASCVD, this information should be added.

Response 1: We thank the Reviewer for this valid coment and we apologize for not showing this important data. We have completed the information in the baseline clinical characteristics of patients in Table 1 and added the definition of cardiovascular disease (CVD) in the “Materials and methods ‑ Patients” section.

“According to the American Heart Association, CVDs were considered as a group of disorders of the heart and blood vessels and included heart disease, heart attack, stroke, heart failure, arrhythmia as also heart valve problems.”

Point 2: Line 134-135. It is not entirely clear what parameter the description “An IVD (in vitro diagnostics) test was utilized, which employed the electrochemiluminescence immunoassay (ECLIA) method” refers to.

Response 2: We thank the Reviewer for this comment and apologize for this ambiguity. The description refers to the IL-6 parameter, so we have made a minor correction that more clearly presents the method of determining the parameter.

“The COBAS e411 immunochemical analyzer (Roche Diagnostics, Warsaw, Poland), which uses an IVD (in vitro diagnostic) test with the electrochemiluminescence immunoassay (ECLIA) method, was used to measure the concentration of serum IL-6.”

Point 3: In table 1 the dimension of BMI should be added.

Response 3: We apologize for this mistake and have added the dimension of BMI in Table 1.

Point 4: I recommend removing the "*" in the tables, you can leave only the description of the statistical methods in captions under the tables.

Response 4: According to the reviewer’s comment, we have removed the "*" in the tables and have left the description of the statistical methods in the captions under the tables, as you suggested.

“Data analysis was performed using the nonparametric Mann-Whitney U test and Pearson's chi-square test”.

Reviewer 2 Report

With interest, I read the manuscript biomedicines-2441415. In my view, a quite nice draft based on a straightforward study.

I have several comments though:

1.      Why was no control group included, e.g. to validate laboratory measurements?

2.      Were any adjustments for potential confounders made?

3.      Lines 64-65: “Circulating serum levels of Lp(a) are primarily determined by the LPA gene locus encoding apo(a).”. I do not get this sentence. Do you want to say that genetic background is the major factor determining serum Lp(a) levels? Besides, it is enough to write “gene” or “locus”.

4.      Lines 266-267: “In a study by Shufta et al., there was a positive correlation between CRP, IL-6, and Lp(a) levels [42] (PMID: 26543308).”. If the referenced paper report the correlation between CRP and IL-6, it is nothing new. IL-6 is well known to be major regulator of CRP synthesis. Or the sentence you wrote is incorrectly written? Besides, the name of the first author of the referenced paper is wrong (not full).

5.      Please, mention that genetic variants in both in this very IL-6 pathway (PMID: 24717336, 26473826) associate with the AV stenosis clinical status.

6.      Was CRP measured? It is written that high CRP excluded from the study but lower levels of CRP (hsCRP) could tell us much about the AVS (see the references mentioned above). Please, provide those and check for the associations with the analyzed parameters.

7.      Starting from the title, the manuscripts suggest some mechanistic findings. Please, keep in mind that you report associations. Any causality (e.g. “predisposing” -> better “predicting/predictive”) is a speculation. Thus, please tone done throughout the draft.

Only moderate adjustments required.

Author Response

Thank you very much for reviewing our work and for all valid suggestions to improve it. Please, consider our point by point response to all comments while we have modified the revised version of the manuscript accordingly. 

Response to Reviewer 2 Comments      

Point 1: Why was no control group included, e.g. to validate laboratory measurements?

Response 1: We thank the Reviewer for this question. We agree with the fact that the inclusion of a group of healthy people with a tricuspid aortic valve as a control group would have been valuable, however, our goal was to prediction the value of Lp(a) as a risk factor of aortic valve stenosis (AVS) in patients with bicuspid aortic valve (BAV). Therefore, our reference group consisted of BAV patients without AVS . However, bearing in mind the clinical importance of obtained results in the curent work and limitations related to the small size of the study group, we plan to continue the study with a larger number of patients in which, according to the reviewer's suggestion, we will include a classic control group to validate laboratory measurements.

Point 2: Were any adjustments for potential confounders made?

Response 2: Due to the relatively small study group and non-parametric distribution of the analyzed variables, we did not use statistical methods to eliminate potential confounding factors, but we tried to eliminate them at the stage of planning through the inclusion and exclusion criteria as well as the division of the population into two age groups.

Point 3: Lines 64-65: “Circulating serum levels of Lp(a) are primarily determined by the LPA gene locus encoding apo(a).”. I do not get this sentence. Do you want to say that genetic background is the major factor determining serum Lp(a) levels? Besides, it is enough to write “gene” or “locus”.

Response 3: We are grateful for this valid comment and apologize for this ambiguity. We have made a correction in the text which allows clearly illustrate the importance of genetic background in determination of serum Lp(a) levels.

Genetic background is the major factor determining circulating serum Lp(a) levels, through the LPA gene encoding apo(a)”

We agree that in general the word "gene" is sufficient given the notation that the LPA gene encodes apolipoprotein(a).

Point 4: Lines 266-267: “In a study by Shufta et al., there was a positive correlation between CRP, IL-6, and Lp(a) levels [42] (PMID: 26543308).”. If the referenced paper report the correlation between CRP and IL-6, it is nothing new. IL-6 is well known to be major regulator of CRP synthesis. Or the sentence you wrote is incorrectly written? Besides, the name of the first author of the referenced paper is wrong (not full).

Response 4: Thank you very much for this valuable comments. We agree that the sentence “In a study by Shufta et al., there was a positive correlation between CRP, IL-6, and Lp(a) levels” is misleading and we apologize for it. We want to apologize also for the incomplete name of the first author of the referenced paper. We have made the necessary corrections.

“In the study by Topçiu-Shufta et al., assumed the correlation of inflammation with Lp(a) in hemodialysis patients the significant positive correlations were observed in individuals with high CRP levels (>10mg/L) not only between Lp(a) and CRP but also between Lp(a) and IL-6 concentration [47].”

Point 5: Please, mention that genetic variants in both in this very IL-6 pathway (PMID: 24717336, 26473826) associate with the AV stenosis clinical status.

Response 5: We are grateful for drawing our attention to this important aspect related to inflammation and the development of AVS. Many thanks for the valuable publication sources, the use of which has helped to improve the value of our work. We have changed part of the “discussion” section.

“Similar to the pathomechanism of atherosclerosis, many studies have pointed to the importance of pro-inflammatory processes in the pathomechanism of AVS development. CRP and IL-6 as the markers of inflammation have been widely studied and described as factors in the development of atherosclerosis. Some works have found a positive association between levels of plasma IL-6 with the progression of AVS. What’s more, not only the concentration of IL-6, but also the genetic variants of this parameter and its receptor seem to be important in the clinical status of AVS [41,42]. The LPA gene contains an IL-6 response element [43] and some data support a positive association between IL-6 and Lp(a). It has been claimed that Lp(a) levels are elevated in patients with increased serum IL-6, which is most closely related to an increased risk of atherosclerosis [22]. The link between CRP and AVS development is not clear. Some researchers have suggested no association [44,45], but others have suggested an association between CRP levels and the incidence of AVS even in a population with low CRP levels [46].”

Point 6: Was CRP measured? It is written that high CRP excluded from the study but lower levels of CRP (hsCRP) could tell us much about the AVS (see the references mentioned above). Please, provide those and check for the associations with the analyzed parameters.

Response 6: Thank you very much for your comment regarding the association of low CRP values with AVS clinical status. All of our patients had CRP measured and we have supplemented the results with the values of this parameter in the table 3 and results of correlations. We have also completed part of the discussion.

Result section: “Interestingly, in our study, there was no significant correlation neither between IL-6 and Lp(a) nor between CRP and Lp(a) in any of the analyzed groups of BAV patients”.

Discussion section:The link between CRP and AVS development is not clear. Some researchers have suggested no association [44,45], but others have suggested an association between CRP levels and the incidence of AVS even in a population with low CRP levels [46]. In the study by Topçiu-Shufta et al., assumed the correlation of inflammation with Lp(a) in hemodialysis patients the significant positive correlations were observed in individuals with high CRP levels (>10mg/L) not only between Lp(a) and CRP but also between Lp(a) and IL-6 concentration [47]. In our study, we showed no significant correlation neither between IL-6 and Lp(a) nor between CRP and Lp(a) in any of the analyzed groups of BAV patients; however, our study did not include patients with acute inflammation (CRP>10 mg/dL) according to the exclusion criteria.”

Point 7: Starting from the title, the manuscripts suggest some mechanistic findings. Please, keep in mind that you report associations. Any causality (e.g. “predisposing” -> better “predicting/predictive”) is a speculation. Thus, please tone done throughout the draft.

Response 7: We thank the Reviewer for this valid suggestion. We agree that our work reports only associations and speculations on causation. Therefore, we have changed the tone in the title and throughout the draft to that Lp(a) may only be a "predictive" factor of earlier AVR.

Round 2

Reviewer 2 Report

Thank you for addressing my comments well.

Minor amendments.